# Objective and Subjective Sleep Measures Are Related to Suicidal Ideation and Are Transdiagnostic Features of Major Depressive Disorder and Social Anxiety Disorder

**DOI:** 10.3390/brainsci13020288

**Published:** 2023-02-08

**Authors:** Heide Klumpp, Fini Chang, Brian W. Bauer, Helen J. Burgess

**Affiliations:** 1Department of Psychiatry, University of Illinois at Chicago, Chicago, IL 60612, USA; 2Department of Psychology, University of Georgia, Athens, GA 30602, USA; 3Department of Psychiatry, University of Michigan, Ann Arbor, MI 48109, USA

**Keywords:** sleep, actigraph, social anxiety, depression, suicidal ideation, suicide

## Abstract

Suicide is a major public health problem and previous studies in major depression and anxiety show problematic sleep is a risk factor for suicidal ideation (SI). However, less is known about sleep and SI in social anxiety disorder (SAD), despite the pervasiveness of SAD. Therefore, the current study comprised participants with major depressive disorder (MDD) (without comorbid SAD) (*n* = 26) and participants with SAD (without comorbid MDD) (*n* = 41). Wrist actigraphy was used to estimate sleep duration, wake after sleep onset, and sleep efficiency; sleep quality was evaluated with self-report. Self-report was also used to examine SI. These measures were submitted to independent *t*-tests and multiple regression analysis. *t*-test results revealed sleep and SI did not differ between MDD and SAD groups. Multiple regression results showed shorter sleep duration and worse sleep quality related to greater SI when taking symptom severity and age into account. Post-hoc partial correlational analysis showed these sleep–SI relationships remained significant after controlling for symptom severity and age. Preliminary findings indicate sleep and SI may be transdiagnostic features of MDD and SAD. Evidence of distinct sleep–SI relationships are consistent with previous reports showing that sleep difficulties contribute to SI. Altogether, improving sleep duration and sleep quality may reduce the risk of SI.

## 1. Introduction

Major depressive disorder (MDD) and social anxiety disorder (SAD) are pervasive and debilitating internalizing psychopathologies in the U.S., with lifetime prevalence rates estimated at 16.6% and 12.1%, respectively [1]. MDD is predominately characterized by sadness and/or anhedonia, whereas SAD is characterized by excessive fear and/or avoidance of situations involving potential scrutiny [2], however, both disorders are associated with significant personal and societal burden. For example, depression imposes a high burden in terms of cost, morbidity, and mortality [3,4,5], and since interpersonal interactions are a part of most facets of daily life, SAD substantially undermines educational attainment, employment, and relationships [6,7,8]. Therefore, an increased understanding of these disorders has the potential to reduce burden.

Accumulating data suggest important clinical features cut across MDD and SAD. Specifically, accruing data in studies of internalizing psychopathologies indicate that problematic sleep (e.g., difficulty falling or staying asleep) is transdiagnostic, though much of this work is based on depression [9,10,11,12,13] and various anxiety disorders (e.g., panic disorder, generalized anxiety disorder) as opposed to SAD specifically [14,15,16,17]. Thus, while sleep difficulties are considered transdiagnostic, this has yet to be directly examined between individuals with MDD and those with SAD. Findings have implications for treatment. For example, psychotherapy studies show treatment improves insomnia [18,19,20,21,22] and behavioral studies show sleep can be extended in individuals with short sleep duration [23,24,25]. Thus, sleep is a putative treatment target.

Over 700,000 people die by suicide each year and, critically, sleep is a risk factor [26]. Evidence that problematic sleep plays a role in suicide includes a meta-analytic study of suicidality in depression that showed depressed individuals with sleep disorders (e.g., insomnia, poor sleep quality, hypersomnia) were more likely to exhibit suicidal ideation, attempt suicide, and complete suicide than those without problematic sleep [27]. Evidence of sleep difficulties being a risk factor for suicidality outside of depression includes a meta-analysis that showed individuals with sleep disturbances had a significantly higher incidence of suicidality (e.g., suicidal ideation, suicide plans, suicide attempts) than those without sleep disturbances [28]. Furthermore, a meta-analysis of longitudinal studies involving clinical and non-clinical cohorts revealed sleep disturbances significantly predicted suicidal ideation, attempts, and death [29]. Links between problematic sleep and suicidality have also been observed in adolescents [30,31]. Moreover, in young adults at risk for suicide, greater sleep disturbance assessed with wrist actigraphy and self-report was found to predict acute increases in suicidal ideation independent of depressed mood [32]. Collectively, problematic sleep is a risk factor for SI in individuals with or without depression.

Beyond sleep, suicidality is also a potential transdiagnostic feature of MDD and SAD. For example, in a National Comorbidity Survey study, 34.8% of individuals with SAD reported suicidal ideation (SI) and associations between SAD and SI have been observed even when controlling for depression [33,34]. With regard to MDD, a recent meta-analysis showed the overall prevalence of SI was 37.7% [35]. Again, though SI is expected to cut across MDD and SAD, this has yet to be tested directly.

In summary, accruing data suggests sleep is a risk factor for SI and individuals with MDD or SAD frequently experience problematic sleep and SI. Therefore, it is important to understand the contribution of sleep to SI in these disorders. Accordingly, the current study evaluated both objective sleep with wrist actigraphy and self-reported sleep quality in addition to SI in participants with MDD or SAD. First, we tested whether overall sleep or SI differed between diagnostic groups (i.e., MDD versus SAD). Because we are not aware of studies that made this head-to-head comparison, we did not have a specific hypothesis as to whether these groups would significantly differ in sleep or level of SI. Secondly, we examined whether individual differences in sleep were uniquely related to SI; we expected worse sleep would correspond with greater SI, regardless of depression or social anxiety severity.

## 2. Materials and Methods

### 2.1. Participants

This is a secondary data analysis from a clinical trial that examined brain-based predictors and mechanisms of psychotherapy in unmedicated, treatment-seeking patients with MDD or SAD (ClinicalTrials.gov Identifier: NCT03175068). The study had no sleep aims; however, participants were provided an opportunity to wear an actigraph device after consenting to the study but before starting treatment prior to COVID-19-based shutdowns. Thus, participants were not preselected for sleep difficulties. Participants were also not preselected for suicidal ideation and the current study is focused on pre-treatment data. All participants were required to be between 18 and 65 years old; exclusionary criteria included: (1) history or current psychosis (e.g., bipolar disorder, schizophrenia); (2) major medical illness with medical history reviewed by a board-certified physician; (3) current active suicidal ideation (i.e., endorse plan or intent) or self-harming behavior; (4) use of psychotropic medication in the last 6 weeks before study entry or during the study; (5) cognitive dysfunction (e.g., traumatic brain injury, dementia, intellectual disability); (6) pervasive developmental disorder (e.g., autism, learning disability); and (7) moderate to severe substance abuse or dependence in the last 6 months. Comorbidity was permitted; however, participants with MDD could not have comorbid SAD and vice versa. See Table 1 for comorbid diagnoses.

Participants were recruited from the community and outpatient clinics via flyers and ads in social media. Interested participants completed a brief phone screening. After the phone screening, potential eligible participants were invited to join a psychiatric interview. The interview comprised clinical measures to determine principal diagnosis and comorbidity. In addition to assessment measures, a Best-Estimate/Consensus Panel of at least three study staff members (e.g., non-treating doctoral-level clinical psychologist, licensed psychiatrist, research assistant) determined study eligibility. All study procedures were conducted at the University of Illinois in Chicago, were approved by the university’s Institutional Review Board and complied with the Helsinki Declaration. All participants were compensated for their time at the rate of $15 per hour for all portions of the study, including the psychiatric evaluation and related assessments. Participants who opted to wear the actigraph device and complete simultaneous sleep diaries were compensated $15 in total from departmental funds.

### 2.2. Measures

#### 2.2.1. Clinical Measures

After obtaining written consent, participants completed the Structured Clinical Interview for DSM-5 [36], Liebowitz Social Anxiety Scale (LSAS) [37], and Hamilton Depression Rating Scale (HAMD) [38] to assess severity of social anxiety and depression, respectively. The rationale for using LSAS and HAMD is that they are interviewer-based assessments, where the LSAS consists of 24 items that assess anxiety/fear and avoidance regarding social and performance situations, whereas the HAMD consists of 17 items that assess depressive, insomnia, and somatic symptoms. Also, cut points permit interpretation of illness severity. Specifically, for the LSAS total score: 0–29 = non-clinical social anxiety, 30–49 = mild social anxiety, 50–64 = moderate social anxiety, 65–79 = marked social anxiety, 80–94 = severe social anxiety, and greater than 95 = very severe social anxiety [37]. For HAMD total scores: below 7 = no depression, 7–17 = mild depression, 18–24 = moderate depression, and ≥25 = severe depression [38]. All interviewer-based measures were conducted by trained staff members. Staff members performing assessments were required to have at least an undergraduate degree in psychology and received training and supervision in conducting clinical assessments, including the LSAS and HAMD, by a licensed doctoral-level clinical psychologist trained in the assessment of social anxiety and depression with over a decade of assessment experience. To evaluate inter-rater reliability, percent agreement based on 10% of randomly selected measures between two trained raters was calculated; results showed percent agreement was 89% for LSAS and 84% for HAMD.

SI was examined with the self-report Inventory of Depression and Anxiety Symptoms, Second Version (IDAS-II) [39] suicidality subscale (IDAS-SS). The subscale involves a Likert-type scale (1 = ‘Not at all’, 5 = ‘Extremely’) and includes six items in total comprising suicidal thoughts (e.g., ‘I had thoughts of suicide’, ‘I thought that the world would be better off without me’) and thoughts of self-harm (e.g., ‘I thought about hurting myself’). The IDAS-SS has been used in outpatient [40] and community [41] samples and has been shown to have good convergent and discriminant validity [39]. Higher scores reflect greater SI. The IDAS-SS in the current study had sufficient internal consistency (Cronbach’s alpha = 0.68).

#### 2.2.2. Sleep Measures

Objective sleep was estimated with wrist actigraphy. Participants were instructed to wear an actigraph device (30-s epochs; Actiwatch Spectrum, Respironics, Bend, OR, USA) on their non-dominant wrist for 7 days/7 nights continuously, press the event marker on the device before and after sleep, and complete simultaneous sleep diaries to inform actigraphy data processing. Data was analyzed with the Actiware 6.0.9 Respironics program. Default software settings were used (10 immobile or mobile minutes for sleep onset or offset and a wake activity count threshold of 40) combined with a standardized approach to finalize the setting of nightly rest intervals, which was guided by event markers, sleep diaries, light data, and activity levels [42]. The following validated sleep variables were used [42]: total sleep time (TST; number of minutes scored as sleep in each rest interval), wake after sleep onset (WASO; number of minutes of all wake epochs between sleep onset and offset), and sleep efficiency (proportion of time from sleep onset to offset in each rest interval scored as sleep). Sleep onset latency was also collected; however, it was not analyzed due to its reduced reliability [43]. All variables were scored for each 24-h period and means were computed.

For subjective sleep, the Pittsburgh Sleep Quality Index (PSQI) [44] was used. The PSQI assesses habitual sleep over the period of a month and consists of 19 questions used to evaluate 7 different components of sleep; all components are summed to yield a global measure of sleep quality. The PSQI has acceptable psychometric properties [45,46] and higher PSQI global scores reflect worse sleep quality. A global score greater than 5 denotes clinically problematic sleep, which is diagnostically sensitive (89.6%) and specific (86.5%) [44]. The PSQI in the current study had acceptable internal consistency (Cronbach’s alpha = 0.70).

### 2.3. Statistical Analyses

A series of independent *t*-tests and chi-square analyses were used to evaluate anticipated and potential differences in clinical measures, sleep measures, and demographic characteristics between MDD and SAD groups. To test for expected distinct relationships between sleep measures and SI (IDAS-SS) when taking symptom severity into consideration, multiple regression analysis (simultaneous entry) was performed with bootstrapping (based on 1000 samples) to evaluate the stability of results. To maximize variance, all independent variables were submitted to one model where the dependent variable (DV) consisted of SI (IDAS-SS). Independent variables (IVs) comprised actigraphy-derived sleep measures (TST, WASO, sleep efficiency), self-reported sleep quality (PSQI), depression (HAMD), and social anxiety (LSAS). All IVs were mean centered. For collinearity to be acceptable, tolerance was required to be >0.20 [47,48]. For significant sleep-related IVs, post-hoc partial correlation analysis was performed to determine if the sleep–SI relationship was maintained when controlling for symptom severity (HAMD, LSAS).

All analyses were two-tailed with alpha level set to 0.05 and performed in the Statistical Package for the Social Sciences (SPSS; Chicago, IL, USA, version 27).

## 3. Results

As anticipated, the MDD group (*n* = 26) was more depressed (HAMD) than the SAD group (*n* = 41) [*t* (65) = 2.90, *p* = 0.005] and the SAD group was more socially anxious (LSAS) than the MDD group [*t* (65) = 10.97, *p* < 0.001]. Yet, despite significant group differences, the average level of depression in the MDD group was in the mild range, whereas the average level of social anxiety in the SAD group was in the severe range [37,38]. Cutting across diagnostic groups, the majority of participants endorsed SI (IDAS-SS) (*n* = 60, 89.6%) and the groups did not significantly differ in level of SI [*t* (65) = 0.79, *p* = 0.43]. The average number of days the actigraph device was worn was 7.82 (*SD* = 2.00) and the average number of sleep diaries completed was 71.6%. Groups did not significantly differ in the number of days the device was worn [*t* (65) = 0.08, *p* = 0.93] or in completion of sleep diaries [χ^2^ (1) = 3.52, *p* = 0.06]. Regarding sleep measures, sleep quality (PSQI) was not collected for one participant due to human error. Neither actigraphic estimates of sleep (TST, WASO, sleep efficiency) nor subjective sleep quality (PSQI) significantly differed between diagnostic groups (lowest *p* = 0.40). See Table 1 for details.

Thus, to aid in characterizing sleep in the current sample, we collapsed across MDD and SAD groups to report the following averages and standard deviations for actigraphic variables as follows: TST = 6.86 h (*SD* = 0.92), WASO = 39.14 min (*SD* = 13.11), and sleep efficiency = 91.38% (*SD* = 2.43). For subjective sleep quality (PSQI global score), the average was 7.87 (*SD* = 3.31), with 74.2% (*n* = 49) meeting the criteria for clinically problematic sleep (i.e., PSQI global score > 5) [44]. Also, even though participants were not preselected for insomnia or hypersomnia, 28.4% (*n* = 19) met DSM-5 [36] criteria for insomnia disorder and 11.9% (*n* = 8) for hypersomnia disorder.

Concerning demographic characteristics, the average age in years across participants was 26.05 (*SD* = 8.16) and the MDD group was older (*M* = 28.61, *SD* = 9.31) than the SAD group (*M* = 24.43, *SD* = 6.98) [*t* (65) = 2.09, *p* = 0.04]. However, the groups did not differ in self-identified gender [χ^2^ (2) = 1.22, *p* = 0.54]; one participant did not identify as male or female. There was also no significant group difference for ethnicity [χ^2^ (1) = 0.05, *p* = 0.82] or race [χ^2^ (5) = 6.02, *p* = 0.30]. See Table 1 for details.

Multiple regression analysis (simultaneous entry) showed collinearity was not acceptable when WASO and sleep efficiency were in the same model (i.e., tolerance < 0.20) and a Pearson correlation analysis verified a strong relationship between these variables (*r* = −0.91, *p* < 0.001); therefore, separate analyses were performed for WASO and sleep efficiency. Also, since the MDD group was older than the SAD group, age was included as an independent variable (IV). Therefore, for one model, IVs consisted of age, depression (HAMD), social anxiety (LSAS), actigraphic variables (TST, WASO), and PSQI (all mean centered), with SI as the dependent variable (DV). The analysis revealed collinearity was acceptable (lowest tolerance = 0.66). Bootstrapped results showed less actigraphic TST (B = −1.72, *s.e*. = 0.57, *p *= 0.006), worse sleep quality (PSQI) (B = 0.67, *s.e*. = 0.23, *p *= 0.005), more depression (B = 0.51, *s.e*. = 0.13, *p *= 0.001), and older age (B = 0.17, *s.e*. = 0.08, *p *= 0.04) related to SI. In contrast, neither WASO (B = −0.01, *s.e*. = 0.04, *p *= 0.79) nor social anxiety (B = 0.03, *s.e*. = 0.02, *p *= 0.14) were significant. Even so, the model was significant [*R*^2^ = 0.73, *F* (6,60) = 11.87, *p *< 0.001].

With regard to the depression finding, it is possible that items in the HAMD that assess for suicide (Item 3) and insomnia (Items 4–6) may have contributed to its significant relationship with SI. Therefore, we re-ran the analysis without these items (i.e., modified HAMD). Specifically, as it was already determined that actigraphy-based TST, subjective PSQI, and age corresponded with SI (IDAS-SS), post-hoc multiple regression analysis was performed with only these IVs along with the modified HAMD (all mean centered). Bootstrapped results showed the modified HAMD did not correspond with SI (B = 0.19, *s.e*. = 0.15, *p *= 0.19), whereas TST (B = −1.21, *s.e*. = 0.56, *p *= 0.03), PSQI (B = 0.88, *s.e*. = 0.21, *p *= 0.001), and age (B = 0.17, *s.e*. = 0.09, *p *= 0.04) each significantly corresponded with SI. The modified HAMD was also not significant when only TST and PSQI were in the model (B = 0.19, *s.e*. = 0.16, *p *= 0.21); again, both sleep variables were significant (lowest *p* = 0.01). Thus, the original HAMD finding appeared to be confounded by links between items for insomnia and SI, with SI as the DV.

To determine whether the significant sleep–SI relationships detected in the multiple regression analysis were distinct, post-hoc partial correlation analysis was performed. Results showed the negative actigraphic TST–SI relationship was maintained when controlling for depression (original HAMD), social anxiety (LSAS), and age (*r* = −0.36, *p* = 0.003). The positive PSQI–SI association was also maintained when controlling for depression, social anxiety, and age (*r* = 0.55, *p* < 0.001). See Figure 1 for scatterplots.

In the model where actigraphic sleep efficiency replaced WASO, bootstrapped results showed that less TST (B = −1.72, *s.e*. = 0.54, *p *= 0.003), worse sleep quality (PSQI) (B = 0.67, *s.e*. = 0.22, *p *= 0.005), more depression (B = 0.51, *s.e*. = 0.14, *p *= 0.002), and older age (B = 0.17, *s.e*. = 0.08, *p *= 0.04) corresponded with SI. However, sleep efficiency (B = 0.06, *s.e*. = 0.20, *p *= 0.74) and social anxiety (B = 0.03, *s.e*. = 0.01, *p *= 0.13) were not significant. Again, the model was significant [*R*^2^ = 0.73, *F* (6,60) = 11.88, *p *< 0.001]. Since results for sleep were the same as the former model, it was not necessary to perform post-hoc regression analysis with the modified HAMD or post-hoc partial correlation analyses.

## 4. Discussion

The current study evaluated whether sleep and SI differed between individuals diagnosed with major depressive disorder (MDD) or social anxiety disorder (SAD) and whether worse sleep corresponded with greater SI regardless of depression and social anxiety severity. Results showed that sleep estimated with wrist-actigraphy and subjective sleep quality did not differ between MDD (without comorbid SAD) and SAD (without comorbid MDD). Similarly, the groups did not differ in SI level. As for associations between sleep and SI, the hypothesis was partially supported as shorter sleep duration (i.e., less total sleep time) assessed with actigraphy, but not wake after sleep onset (WASO) or sleep efficiency, corresponded with greater SI. Also, worse self-perceived sleep quality corresponded with more SI. Importantly, these sleep–SI relationships were maintained when controlling for symptom severity, indicating depression and social anxiety did not drive the relationships.

Altogether, results extend previous study findings that indicate sleep [9,10,11,12,13,14,15,16,17] and SI [33,34,35] cut across categorical diagnostic boundaries. Findings also highlight the importance of assessing sleep and SI in SAD, which is frequently under-recognized in clinical settings [49] and under-researched in studies of sleep and suicidality compared to depression. Concerning sleep, it is not possible to determine whether actigraphic estimates of sleep or subjective sleep quality were abnormal, as there was no comparator healthy control group in the study. Even so, the majority of participants met the criteria for clinically problematic sleep based on self-report (i.e., PSQI) [44].

Though determination of atypical sleep was not a focus of the study, it is notable that average sleep duration indexed with actigraphy was 6.8 ± 0.92 h a night, which suggests many of the participants did not obtain the recommended 7 h of sleep a night for adults [50]. Since insufficient sleep is associated with SI in the general population [51] and other negative outcomes such as impairment in executive function [52,53], risk-taking behavior [54], and common health problems (e.g., hypertension, diabetes) [55], extending sleep in MDD and SAD has the potential to make an important public health impact. Evidence that sleep can be extended includes behavioral interventions for individuals with short sleep duration [23,24,25].

With regard to the association between objective estimates of sleep and SI, multiple regression results were significant for sleep duration where shorter duration (i.e., less total sleep time) related with more SI when taking symptom severity and age into account and when controlling for these variables via partial correlation. Yet, regression results were not significant for WASO or sleep efficiency. Results are consistent with previous studies. For example, in a meta-analysis involving objective estimates of sleep, patients with SI were found to have significantly shorter sleep duration than those without SI [56], while in an ecological momentary assessment study, less actigraphy-based total sleep time was associated with higher levels of next-day SI in participants with SI [57]. Yet, actigraphy-based sleep efficiency and sleep onset latency did not predict SI (WASO was not examined) [57]. Accruing findings suggest that among sleep parameters, sleep duration may be a biomarker of SI. However, in light of gaps in the literature and a dearth of studies that have examined wrist-actigraphy sleep parameters and SI in MDD or SAD, it is important to replicate results in a large sample before drawing firm conclusions. As expected, worse subjective sleep quality also significantly corresponded with greater SI when taking symptom severity and age into consideration and when controlling for these variables. Taken together, shorter sleep duration and worse self-perceived sleep quality were each distinctly associated with greater SI.

Suicidality is complex and the role of sleep in SI is likely to involve various pathways. One potential pathway may involve problems regulating negative emotions. For example, cognitive impairment associated with sleep loss (i.e., sleep-related ‘hypofrontality’) is thought to contribute to the greater likelihood of suicide at night relative to other times of the day [58]. Moreover, neuroimaging studies that point to shared sleep and emotion regulation systems [59] and evidence of atypical neurofunctional activity in regions central to emotion processing and regulation (e.g., amygdala, anterior cingulate cortex, prefrontal cortices) in depression and anxiety disorders [60,61] suggest problematic sleep may exacerbate dysfunction, which increases the risk of SI and suicidal behavior. In further support of the role sleep plays in emotion and regulation, laboratory studies involving healthy participants show insufficient sleep modulates brain regions (i.e., increases amygdala response) and circuitry (prefrontal-amygdala connectivity) when viewing negative stimuli [62] and disrupts the ability to implement an adaptive emotion regulation strategy (i.e., reappraisal) in the context of negative information [63]. Since the delineation of sleep–SI mechanisms is beyond the scope of the current study, it will be important for future studies to test the role sleep has on SI in MDD and SAD using an emotion regulation framework.

Multiple regression analysis also showed that greater depression assessed with HAMD, but not social anxiety evaluated with LSAS, corresponded with SI when taking sleep into account. In contrast to the HAMD, the LSAS does not assess for insomnia or suicidality; when items that assessed for these were removed, the HAMD no longer significantly corresponded with SI. Therefore, depression symptoms outside of insomnia and suicidality (e.g., depressed mood, feelings of guilt) did not appear to relate to SI when taking actigraphy-based estimates of sleep and subjective sleep quality into consideration. Age also positively corresponded with SI. However, since the average age across participants was relatively young (i.e., 26 years), we hesitate to interpret this finding and speculate that factors associated with SI such as work-related stress [64], loneliness [65], and disability [66] may have contributed to the finding. Since stressors and other challenges differ across the age spectrum, further study is needed to understand factors that contribute to the positive age–SI relationship.

### Limitations

Findings need to be considered in the context of important limitations. First, the sample sizes were small and results may not generalize to individuals who are receiving pharmacotherapy or who differ in demographic or clinical characteristics, including those with active SI or recent history of self-injurious behavior or suicide attempt. Second, the severity of depression in the MDD group was in the mild range, which may have reduced our ability to detect depression-related effects. Third, since there was no healthy control group, it was not possible to determine whether estimates of sleep were atypical or whether any other measure, including use of the actigraph device, was aberrant. Fourth, age significantly differed between diagnostic groups; therefore, we cannot rule out the possibility that age influenced findings. Fifth, the cross-sectional design does not permit evaluation of sleep as a predictor of SI. Sixth, we did not screen for certain sleep disorders (e.g., obstructive sleep apnea) or measure or instruct participants to refrain from substances that may impact sleep (e.g., caffeine, alcohol), which may have introduced confounds. Seventh, a portion of participants (28.4%) did not complete a sleep diary; hence, the event marker press reflecting sleep onset/offset could not be confirmed in these participants, and we could not verify that participants adhered to instructions on completing sleep diaries. Eighth, relatively few participants met the full criteria for insomnia or hypersomnia disorders, yet the majority met the criteria for clinically problematic sleep based on the PSQI; therefore, results are more consistent with a dimensional than categorical nosology of problematic sleep. Ninth, we did not collect data on nightmares or excessive daytime sleepiness, which are also risk factors for suicide [30,67,68,69]. Tenth, there was no direct manipulation of sleep (i.e., sleep deprivation in a controlled setting); therefore, results rely on estimates of naturalistic sleep. Eleventh, there were more participants with SAD than with MDD and data collection was interrupted by COVID-19-related shutdowns. Therefore, it will be important to replicate results in a large sample where there is an equal number of participants in each diagnostic group. Lastly, data on participants’ weekday/weekend schedules or preferred times to engage in activities or sleep (e.g., chronotype) was not obtained; therefore, we cannot rule out the possibility that they may have impacted findings.

## 5. Conclusions

Preliminary results suggest objective and subjective indices of sleep and SI cut across SAD and MDD. Also, evidence that shorter actigraphic sleep duration and worse self-perceived sleep quality each corresponded with SI even when controlling for depression, social anxiety, and age indicates these sleep parameters may uniquely contribute to SI. Finally, multiple regression results showed age and depression positively related to SI when taking social anxiety and sleep measures into consideration. However, participants were relatively young, the level of depression was in the mild range, and the depression finding was confounded by items that assessed for sleep and suicidality. Therefore, we hesitate to interpret these results. Despite limitations, the significant sleep–SI findings have important implications for treatment; for example, extending sleep and improving sleep quality may reduce the risk of suicidal ideation.

## Figures and Tables

**Figure 1 brainsci-13-00288-f001:**
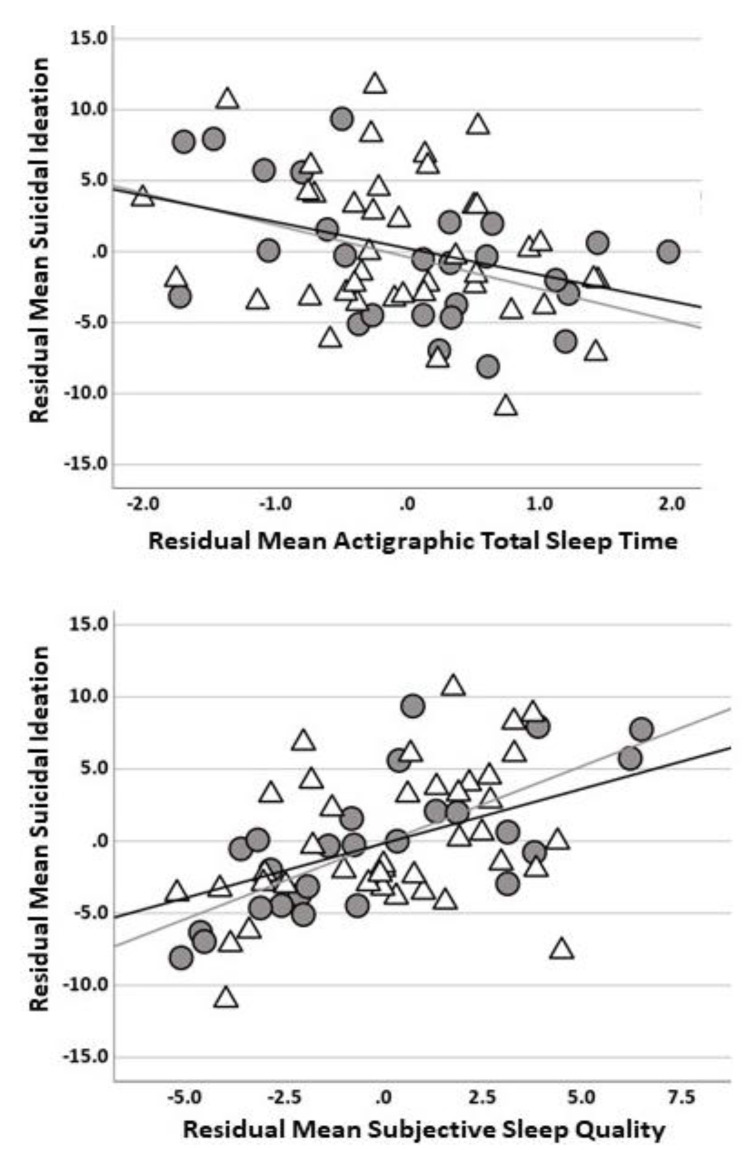
Scatterplot illustrating relationship between actigraphic total sleep time and suicidal ideation controlling for depression (HAMD), anxiety (LSAS), and age in years (i.e., residuals) (top panel); Scatterplot depicting relationship between sleep quality (PSQI) and suicidal ideation controlling for depression (HAMD), anxiety (LSAS), and age in years (i.e., residuals) (bottom panel). Note: Suicidal ideation = Inventory of Depression and Anxiety Symptoms, Second Version suicidality subscale; HAMD = Hamilton Depression Rating Scale; LSAS = Liebowitz Social Anxiety Scale; PSQI = Pittsburg Sleep Quality Index; gray circles and gray fit line = major depressive disorder; black triangles and black fit line = social anxiety disorder.

**Table 1 brainsci-13-00288-t001:** Clinical, sleep, and demographic characteristics for diagnostic groups: Values reflect the mean, and standard devations are in parentheses.

	MDD (*n* = 26)	SAD (*n* = 41)
Hamilton Depression Rating Score	11.30 (3.77)	8.34 (4.24)
Liebowitz Social Anxiety Score	36.76 (15.9)	83.68 (17.73)
IDAS-II Suicidality Subscale	14.80 (6.66)	13.58 (5.79)
Actigraphic total sleep time (hours)	6.84 (1.03)	6.88 (0.86)
Actigraphic wake after sleep onset (minutes)	38.10 (14.86)	39.80 (12.01)
Actigraphic sleep efficiency (%)	91.51 (2.57)	91.29 (2.38)
Pittsburgh Sleep Quality Index global score	8.30 (3.67)	7.60 (3.07)
Age in years	28.61 (9.31)	24.43 (6.98)
Gender (%)	%	%
Female	69.2	58.5
Male	30.8	39.0
Not Reported	0.0	2.4
Ethnicity (% Hispanic/Latinx)	19.2	17.1
Racial identity (%)	%	%
White	46.2	48.8
Black	15.4	17.1
Asian	15.4	26.8
More than one race	11.5	2.4
Other	11.5	2.4
Not reported	0.0	2.4
Comorbid diagnoses (%)	%	%
Generalized anxiety disorder	46.2	51.2
Insomnia	38.5	22.2
Hypersomnolence	11.5	12.2
Specific Phobia	7.7	9.8
Persistent Depressive Disorder	30.8	0.0
Panic Disorder	0.0	7.3
Post-traumatic Stress Disorder	0.0	2.4
Attention-Deficit/Hyperactivity Disorder	0.0	4.9
Adjustment Disorder	0.0	2.4

Note: SAD = Social Anxiety Disorder; MDD = Major Depressive Disorder; IDAS-II = Inventory of Depression and Anxiety Symptoms, Second Version.

## Data Availability

The data and material that support the findings of this study are available from the corresponding author upon reasonable request.

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
