# Peer review of "Objective and Subjective Sleep Measures Are Related to Suicidal Ideation and Are Transdiagnostic Features of Major Depressive Disorder and Social Anxiety Disorder"

_brainsci, 2023, doi:10.3390/brainsci13020288_

Round 1

Reviewer 1 Report

The authors present a study exploring differences in sleep and suicidal ideation between a group of participants with major depressive disorder (MDD) and a group of participants with social anxiety disorder (SAD). The study also explores how objective and subjective sleep measures, depression and social anxiety scores, and age predict suicidal ideation. Despite the various limitations identified in the study by the authors themselves, and the preliminary nature of the findings, the results are interesting and contribute positively to the body of knowledge linking sleep difficulties and suicidal ideation, and suggest that sleep problems and suicidal ideation might be common features to MDD and SAD.

I would like to make the following suggestions for improvement of the manuscript:

1.       I feel that the title of the article does not provide a clear picture of the study, as the objectives of the work seem to be twofold: 1) compare participants with MDD and participants with SAD on a number of variables, including sleep measures and suicidal ideation, and 2) explore the predictors of suicidal ideation. Perhaps the authors could reformulate the title to reflect better the aims of the study.

2.       Although the authors state that participants from the MDD group could not have comorbid SAD and vice-versa, and the two groups significantly differ in their depression scores on the HAMD (MDD = 11.30; SAD = 8.34), these values sound somehow strange. According to the usual scoring guidelines of the HAMD, on the one hand, the scores of the MDD group seem considerably low, as only a score higher than 20 is usually considered to be indicative of moderate depression (much higher than the reported for the MDD group). On the other hand, only scores up to 7 are considered to be in the normal range/no depression (whereas the score reported for the SAD group is higher than 7). Could the authors please clarify? Does this challenge the conclusions that can be drawn from the results on depression scores?

3.       Line 107 - The usual cut-off scores for the LSAS and HAMD scales should be mentioned.

4.       Internal consistency scores (Cronbach's alpha) for the sample of the present study should be mentioned for the IDAS-SS (lines 115-122) and PSQI scales (lines 139-145).

5.       Why were depression scores a significant predictor of suicidal ideation, but social anxiety scores did not predict suicidal ideation? Do the authors have any tentative explanation for this result? This should be discussed. Also, does this finding need to be taken in consideration when claiming the transdiagnostic nature of suicidal ideation, i.e., how does this impact the claim that SI is transdiagnostic?

6.       Lines 255-257 – The authors state that “Findings suggest variance in SI may not be sensitive to individual differences in fragmented sleep (i.e., WASO) or actual time spent asleep while in bed (i.e., sleep efficiency)”, but do not offer any explanation for this finding. Do the authors have any tentative explanatory hypothesis as to why this might be the case? This should be discussed.

7.       Considering the attempt of the study to provide support the transdiagnostic nature of suicidal ideation and sleep problems in MDD and SAD, it would seem important to test if suicidal ideation and the various sleep measures were significant predictors of depression and social anxiety scores in the two study groups separately. Why were these analyses not carried out? Despite the relatively small sample, considering the two groups separately, perhaps these analyses could still be included in the paper, as preliminary findings.

Reviewer 2 Report

Very interesting topic and of great relevance for potential readers of this journal.

However, some minor comments are made in favor of improving the current version of the manuscript.

.- Abstract and introduction, ok.

.- Methodology. Doubt arises. Reference is made to the clinical trial with code NCT03175068. How many patients did that trial include?. Why are 26 and 41 patients selected? and weren't they paired equally, e.g, 41:41?. Why didn't they include a group of healthy patients?

When you say “All participants were compensated for their time”, what do you mean? Maybe they received some sort of bonus? And if so, wouldn't it imply a “selection bias”?

When you say “After obtaining consent”, could “written consent” be specified/added?

Liebowitz Social Anxiety Scale (LSAS) and Hamilton and Depression Rating Scale (HAMD), why these scales and not others? Briefly argue.

Trained staff members”, who? Nurses, social workers or others? Could you specify succinctly?

.- Results. The high level of compliance with the use of the nocturnal device in both study groups (>70%) is striking. Likewise, it should be noted that there were no significant differences between the two groups in terms of the number of days the device was used. They also did not differ in the number of suicidal ideations.

.- Discussion. Very improvable. There is a lack of interpretation of the results (see comments on results).

When do you talk about “transdiagnosis”? what is it referring to?

Highlighting the impact of sleep, laboratory studies involving healthy participants show…” is not related to the objective of the study. Assess delete.

Limitations. Low sample of the groups. A control group (healthy) is missing to assess not only atypical sleep estimates, but also non-use of the device, hours of sleep, suicidal ideation and compare the results with the other two groups.

.- Conclusions. Evaluate redoing/deleting the last sentence, since it does not fit the objectives of the study.

.- Bibliography. Of the 62 references provided, only 19 (30.6%) are recent, that is, 5 years or less. Assess including any recent additional reference.

.- Figures and Tables. okay.
